# Roles of Tryptophan and Charged Residues on the Polymorphisms of Amyloids Formed by K-Peptides of Hen Egg White Lysozyme Investigated through Molecular Dynamics Simulations

**DOI:** 10.3390/ijms24032626

**Published:** 2023-01-30

**Authors:** Husnul Fuad Zein, Thana Sutthibutpong

**Affiliations:** 1Nanoscience and Nanotechnology Program, Faculty of Science, King Mongkut’s University of Technology Thonburi (KMUTT), 126 Pracha Uthit Rd., Bang Mod, Thung Khru, Bangkok 10140, Thailand; 2Department of Physics, King Mongkut’s University of Technology Thonburi (KMUTT), Bangkok 10140, Thailand; 3Center of Excellence in Theoretical and Computational Science (TaCS-CoE), Faculty of Science, King Mongkut’s University of Technology Thonburi (KMUTT), 126 Pracha Uthit Rd., Bang Mod, Thung Khru, Bangkok 10140, Thailand

**Keywords:** amyloid fibril, hen egg, white lysozyme, molecular dynamics simulations

## Abstract

Atomistic molecular dynamics simulations of amyloid models, consisting of the previously reported STDY-K-peptides and K-peptides from the hen egg white lysozyme (HEWL), were performed to address the effects of charged residues and pH observed in an in vitro study. Simulation results showed that amyloid models with antiparallel configurations possessed greater stability and compactness than those with parallel configurations. Then, peptide chain stretching and ordering were measured through the end-to-end distance and the order parameter, for which the amyloid models consisting of K-peptides and the STDY-K-peptides at pH 2 displayed a higher level of chain stretching and ordering. After that, the molecular mechanics energy decomposition and the radial distribution function (RDF) clearly displayed the importance of Trp62 to the K-peptide and the STDY-K-peptide models at pH 2. Moreover, the results also displayed how the negatively charged Asp52 disrupted the interaction networks and prevented the amyloid formation from STDY-K-peptide at pH 7. Finally, this study provided an insight into the interplay between pH conditions and molecular interactions underlying the formation of amyloid fibrils from short peptides contained within the HEWL. This served as a basis of understanding towards the design of other amyloids for biomaterial applications.

## 1. Introduction

Amyloid fibrils are denatured fibrous forms of protein aggregates that have been intensively studied since their deposition is related to a variety of disorders, including Alzheimer’s and Parkinson’s disease, type II diabetes, and prion diseases [1,2,3,4]. The improved understanding of aggregation processes led to some attempts to prevent or interfere with the fibrillation by adding either stabilizing proteins [5,6], or small molecules [7]. However, the formation of fibrils is not limited to proteins associated with neurodegenerative diseases. Any protein can form amyloid fibrils under certain conditions [8,9,10], and some amyloid fibrils play functional roles for organisms, such as formation of adhesive biofilms [11,12], protective eggshells [13], and secretion mechanisms of some toxic compounds [14,15,16]. In more recent studies, an amyloid fibril was synthesized in vitro as a component of functional biomaterials, due to its chemical stability, biocompatibility, and programmability [17]. Different forms of amyloid-based functional materials, e.g., hydrogels [18], high-strength materials [19], composite materials [20], and conductive materials [21] led to applications in the fields of (i) sensing materials by combination with other sensing components that respond to varying physico-chemical conditions [22,23,24,25,26], (ii) catalytic materials by titrable amino acid residues that facilitated electron–proton transfer [27,28,29], and (iii) scaffolds that mimicked the extracellular matrix environment for cell adhesion and growth [30,31].

Among many possible amyloidogenic proteins, hen egg white lysozyme (HEWL) has been widely used as a model protein for the study of amyloidosis due to its abundance, and convenience for isolation and purification [32,33]. The use of HEWL for amyloid-based functional biomaterials also became more interesting, as several studies seeking optimal physico-chemical conditions for HEWL fibrillation were carried out, along with the combined in vitro and in silico analyses on the mechanisms of HEWL aggregates under simulated physico-chemical environments [34,35]. Understanding the fundamental aspects of stabilization and destabilization of amyloid fibrils from HEWL, or some important peptide fragments from HEWL, might shed some light on the precise control of amyloidosis—either to prevent pathogenic amyloids, or to fabricate parts of functional biomaterials [26].

As a HEWL contains many positively charged amino acids, it was found that polymers or surfactants with anionic functional groups, e.g., sodium dodecyl sulfate (SDS) [36] and heparan sulfate [37], could promote amyloidosis of HEWL, which was confirmed by a computational analysis [38]. Effects of pH and temperature were also investigated as Xing et al. reported that an optimum condition for the formation of HEWL in vitro was at pH 2 and 65 °C [39]. This was later confirmed by Alam et al. [40], and our previous computational study on the effects of heavy protonation at pH 2 to the re-distribution of hydrophobic residues that facilitate the transition into beta amyloids [41]. An earlier report on the low pH amyloid formation from HEWL at 65 °C demonstrated the 49–101 region was amyloidogenic, and the non-amyloidogenic 1–48 and 102–129 regions slowed down formation [42]. Further analysis on the molecular dynamics (MD) simulations of HEWL also showed that the region of residues 54–62, formerly a connecting loop between two small beta sheets buried inside a native HEWL structure, turned into a beta strand under an elevated temperature at pH 2. That 54–62 region containing a ‘GILQINSRW’ sequence was previously reported as the ‘K-peptide’, and was believed to be the core region for the amyloidosis of HEWL [43]. Tokunaga et al., demonstrated the importance of the tryptophan 62 residue, as the variants without tryptophan 62 displayed no sign of fibrillation. It was also found that the longer sequence ‘STDYGILQINSRW’ or STDY-K-peptide also formed amyloids at lower pH [44]. Different pH conditions also caused a bifurcation of the amyloid polymorphisms formed by the whole HEWL, as a long and flexible amyloid was seen at a strongly acidic condition, and a short and rigid amyloid was seen at a weakly acidic condition [45]. However, another observation of the amyloids formed by short peptides isolated from both HEWL (ILQINS) and human lysozymes (ILQINS), within the K-peptide region, showed that the lower pH resulted in a higher propensity of forming amyloid crystals [46]. The complete explanation of these different features of HEWL-based amyloids was still lacking.

Therefore, this study employed molecular simulation techniques and improved analysis protocols on the model systems of 2 × 8 beta amyloids of short peptide sequences from HEWL. The K-peptide amyloid model was considered, along with the STDY-K-peptide models at pH 2 and pH 7, differentiated by the protonation states of aspartic acid (D) residue. Starting structures included both parallel and antiparallel orientations of neighboring peptide pairs to further verify the known conformational stability of antiparallel beta amyloid systems. Detailed conformational analysis, along with the pairwise interactions between the amino acid and the chain order parameters, were performed to provide some insight on the complex interplay between pH conditions and intermolecular interactions.

## 2. Results and Discussions

### 2.1. Disruption of the Antiparallel Beta Amyloid with a Negative Charge Addition

Figure 1a–f displayed the final snapshots after 100 ns of all six simulated beta amyloids (Table 1). A K-peptide GILQINSRW (54–62) contained a positively charged arginine residue (R; labelled in blue), along with four polar residues (labelled in green), and four non-polar residues (labelled in white). Glycine residues (G) were considered as polar residues because the polar amine and carbonyl groups of their backbone were dominant for the interaction network. Meanwhile, a STDY-K-peptide STDYGILQINSRW (50–62) contained a positively charged arginine (R; labelled in blue) and negatively charged aspartic acid residues (D; labelled in red) near both ends, along with seven polar residues, and four non-polar residues. At a highly acidic condition of pH 2, previously investigated by some previous studies on the amyloidosis of HEWL and its constituent peptide, the aspartic acid residue of each peptide was protonated into an uncharged polar residue, and its contribution to electrostatic interactions became weakened.

The simulated ‘antiparallel’ K-peptide system after 100 ns (54–62-AP; Figure 1a) showed that the beta amyloid model of two beta sheets, each consisting of eight beta strands, was changed from the planar starting conformation of beta sheets (grey shadow) to a pair of left-handed twisting sheets. The antiparallel STDY-K-peptide (50–62-AP) in Figure 1b was more deviated from the starting structure, as the interaction network at the middle of upper beta sheet was lost. It could be seen from the deprotonated 50–62-AP model that an alternating configuration of arginine and aspartic acid sidechains could promote a strong electrostatic interaction network between positive and negative monopoles. However, the simulated STDY-K-peptide model at a low pH (50–62-pH2-AP; Figure 1c) changed back into the left-handed twisting sheets when the aspartic acid (D) residues were protonated, and the monopole interactions became only repulsive forces between arginine sidechains. Stability of both positively charged K-peptide, and protonated STDY-K-peptide amyloid models contributed by other interaction terms, will be discussed in later sections.

### 2.2. Antiparallel Beta Amyloid Models Displayed Greater Compactness and Stability

Considering amyloid models with ‘parallel’ configurations, Figure 1d–f displayed the simulated 54–62-P, 50–62-P, and 50–62-pH2-P models after 100 ns. The 54–62-P model in Figure 1d was with all K-peptides oriented in parallel at the start, and all positively charged arginine residues near the C-termini were placed together. The highly positive charge cluster of arginine residues within the 54–62-P system could be neutralized by the negative charges of deprotonated carboxylate groups of C-termini. Meanwhile, the STDY-K-peptide models at both neutral (50–62-P; Figure 1e) and acidic conditions (50–62-pH2-P; Figure 1f) were found with all eight parallel peptide strands within a beta sheet; however, the two beta sheets preferred to be antiparallel, in concurrence with the addition of STDY segments containing negatively charged residues.

Figure 2a–c displayed the root mean square deviation (RMSD) of all simulated amyloid structures relative to the perfectly parallel or antiparallel configurations obtained from rigid docking calculations. RMSD values from all calculations converged after 50 ns, suggesting that equilibriums were reached. The 54–62-AP and 50–62-pH2-AP simulations had the lowest and second lowest RMSD during the last 50 ns, respectively, which were due to beta sheet twisting. A slightly higher RMSD was found for the 50–62-AP simulation due to the upper beta sheet disruption. Figure 2a–c also showed that all simulated parallel amyloid models had a higher RMSD than their antiparallel counterparts. The RMSD of 54–62-P could be attributed to a shear sliding motion between two beta sheets, while the RMSD of both 50–62-P and 50–62-pH2-P could be attributed to the higher amount of conformational transition from beta strands to random coils, for which the order parameter of peptide chains will be discussed in later sections.

Moreover, the compactness of each simulated amyloid system was measured through the radius of gyration (Rg). Rg of the simulated amyloids, consisting of 16 K-peptides, was within the range of 1.5–1.7 nm (Figure 2d), while Rg of the STDY-K-peptide system was within the range of 1.8–1.9 nm (Figure 2e–f). Rg calculations displayed a similar trend to the RMSD calculations, as all amyloid systems with ‘antiparallel’ relative orientation of peptides possessed a smaller Rg than their ‘parallel’ counterparts, signifying the greater compactness of the antiparallel beta amyloid. The higher RMSD and radius of gyration of the parallel amyloid models were also in concurrence with the higher root mean square fluctuation (RMSF) calculated from the last 50 ns of all simulations, presented in Figure 2g–i. RMSF profiles of all six simulations shared some common characteristics, i.e., broad regions of low RMSF around the middle amino acids ILQINS investigated in a previous study [46], and the RMSF maxima found at either N- or C-termini corresponded to the observed beta-to-random coil transition. However, the C-terminal residues of the 54-62-AP and 50–62-pH2-AP systems possessed the lowest RMSF and the highest conformational stability. These C-terminal residues corresponded to Trp62 residues, proposed as a key residue for HEWL amyloidosis.

### 2.3. Antiparallel Amyloid Models of K-Peptides and STDY-K-Peptide at pH 2 Displayed Stretched and Ordered Chains

To further characterize the simulated amyloid models, end-to-end distance was measured for each peptide strand, and order parameter was determined from the orientation vectors defined through the displacement between C-alpha atoms of residues i and i + 2 (Figure 3). Figure 3a showed that the end-to-end distances of most peptide strands within the antiparallel amyloid models were around 0.29–0.32 nm per residue. However, the outermost peptide strands from some sheets became significantly shortened due to a weaker interaction network, compared to the inner strands. Figure 3a also showed that only the two outermost peptides of the 54–62-AP model were found with their end-to-end distance below 0.29 nm per residue, while three and eight shrunken peptides were found for the 50–62-pH2-AP and 50–62-AP systems, respectively. The parallel amyloid models in Figure 3b displayed an even higher number of shrunken peptides when compared with their antiparallel counterparts, which corresponded with elevated levels of conformational distortion and fluctuation, signified by their higher RMSD and RMSF.

However, the shorter end-to-end distance of the parallel amyloids was in concurrence with the larger radius of gyration. The relationship between chain stretching and Rg from the simulated amyloid system contradicted that seen in a single polymer chain, in which Rg increased with polymer chain stretching. This interesting observation from the models consisting of aggregated peptides could be further quantified through the chain order parameter, defined through the relative orientation of displacement vectors between C-alpha atoms of residues i and i + 2. This definition was chosen so that the conformation of a perfect beta strand with a zigzag conformation gave rise to perfectly parallel displacement vectors, and the order parameter of 1. Figure 3c–d displayed the chain order parameter of simulated amyloid systems along the 100 ns trajectories. At the beginning of the simulations, all simulated models displayed their relatively high order parameters obtained from the generated linear conformation of peptides, and the rigid docking calculation in vacuum. As the systems were subjected to the solvent environment and NPT ensemble, conformational relaxation and fluctuation occurred and caused the decrease in the order parameter of all systems. After 50 ns, the order parameter of the 54–62-AP and 50–62-pH2-AP systems became equilibrated around 0.55 ± 0.05, while the 50–62-AP system had a lower order parameter around 0.38 ± 0.02 (Figure 3c).

According to Figure 3d, the order parameters of parallel amyloid models became lower than all the parallel amyloid models. The partial folding of the random coil segments of shrunken peptides within the parallel amyloid models affected the relative chain orientation and lower order parameter. The lower order parameter was reflected by the higher rate of beta-to-coil transition. Partial peptide folding mostly occurred through the outermost amino acids of the outermost peptides increasing the vertical dimension of amyloids, while the longitudinal and lateral dimensions were preserved. Therefore, Rg of the simulated parallel amyloid models was increased, despite shorter end-to-end distance. On the other hand, 54–62-AP and 50–62-pH2-AP systems, with the largest order parameter, could preserve most of the straightened beta strands with high compactness and stability. This in silico observation was consistent with a series of HEWL peptide aggregation experiments by Tokunaga et al. [44], in which aggregated K-peptides could form amyloid fibrils at pH 2, 4, and 7, while the STDY-K-peptide could form even higher amounts of amyloid fibrils at pH 2 and 4, but not at pH 7. Moreover, low RMSF values, that signified the stability of C-termini of both 54–62-AP and 50–62-pH2-AP, also addressed the importance of Trp62 for the amyloidosis of HEWL which was suggested by some other previous studies [47,48,49].

### 2.4. Roles of Trp62 Capping the Ends of Highly Ordered Amyloid Models

To further investigate the molecular mechanisms underlying the stability of the simulated antiparallel amyloid models of the K-peptides and STDY-K-peptides, average molecular mechanics energy contributions of each amino acid to the interaction between the sheets and strands were calculated (Figure 4). Considering the energy decomposition between sheets in Figure 4a, the positive energy found at the N-terminal residues and the Arg61 residues of 54–62-AP and 50–62-pH2-AP systems was due to the electrostatic repulsion between the protonated N-terminal amino groups, and the positively charged arginine sidechains. Positive energy contribution from amino acid residues disappeared in the simulation of the 50–62-AP model, due to the presence of deprotonated Asp52 residue causing additional coulombic attraction between sidechains of Asp52 and Arg61. For each model, Trp62 residues at the C-terminus had the largest energy contributions. This might be due to the large sidechain of tryptophan that can either form hydrophobic contacts or hydrogen bonds with nearby residues. Similar trends were observed for the interactions between neighboring peptide strands (Figure 4b), but with stronger contributions from hydrogen bonding at each amino acid. Arg61 of 54–62-AP and 50–62-pH2-AP systems also displayed a positive energy contribution, while Asp52 and Arg61 even displayed stronger energy contributions than Trp62. However, without the contribution from coulombic attraction between Asp52 and Arg61, 54–62-AP and 50–62-pH2-AP models displayed the strongest contribution of −40 ± 12 kJ/mol by the Trp62 residues.

The roles of Trp62 residues in local interactions within the amyloid structures were further investigated through the radial distribution functions (RDF) between end residues of peptides shown in Figure 5. Figure 5a displays the arrangement of terminal residues within the 54–62-AP model, in which Trp62 (orange) residues at the C-termini are surrounded by Gly54 (green) and Ile55 (white) residues at the N-termini of neighboring peptide strands. The highest first peaks of RDF, between all pairs of Trp62 and all pairs of Gly54 at both termini, were found for the 54–62-AP model, when compared with those for 50–62-AP (Figure 5b) and 50–62-pH2-AP (Figure 5c). The first RDF peaks between Trp62 and the N-terminal residues (Gly54 for 54–62-AP and Ser50 for 50–62-Ap models) were similar for all models, but the second RDF peak could be seen only for the 54–62-AP model, suggesting a higher order of arrangement. Moreover, RDF peaks between Trp62 and the second N-terminal residues of the 54–62-AP model were significantly higher than those of the other two models.

Both RDF profiles and the snapshots in Figure 5 showed that terminal residues of the antiparallel amyloid, consisting of K-peptides, were orderly arranged. The pairing between Trp62 with the largest sidechain, and Gly54 with the smallest sidechain could minimize steric repulsions, while hydrophobic contacts between Trp62 and Ile55 are still maximized. The addition of negatively charged Asp52 residues for the antiparallel amyloid, consisting of STDY-K-peptides, introduced coulombic attractions between Asp52 and Arg61 residues, which strengthened the peptide aggregation but also disrupted the ordering of terminal residues. Protonation of Asp52, within the amyloid consisting of STDY-K-peptides at pH 2, neutralized the negatively charged residues, so that coulombic interactions between monopoles disappeared and interactions contributed by Trp62 became important. The highly ordered terminal residue arrangement, capped by C-terminal Trp62 residues for amyloids consisting of K-peptides and protonated STDY-K-peptides, was in concurrence with their peptide arrangement under high order parameters.

## 3. Materials and Methods

### 3.1. Beta Amyloid Model Creation

Firstly, we prepared atomistic structures for three peptides: (i) K-peptide (GILQINSRW), (ii) STDY-K-peptide (STDYGILQINSRW) and (iii) STDY-K-peptide at pH 2 (STDYGILQINSRW; protonated form), by using the peptide builder tools implemented within the Avogadro 2.0 software (University of Pittsburg, Pittsburgh, PA, USA) to obtain straightened peptides as starting structures [50]. Then, a rigid protein–protein docking was performed in the HDOCK webserver [51] for a pair of peptide models with the same structure to search for the binding modes of the first stage of fibrillation. Binding modes tended to form ‘parallel’ and ‘antiparallel’ beta sheets with the lowest binding energy scores, which were kept for the second rigid protein–protein docking between two peptide pairs to create the peptide quartet. After that, the third and the fourth docking calculations were performed to create the systems of eight and sixteen peptides, respectively. For all three peptides, the amyloid models obtained with the lowest possible binding energy score for each rigid docking calculation resulted in the sixteen-peptide systems of two beta sheets, each consisting of eight beta strands.

### 3.2. Atomistic Molecular Dynamics Simulations

Six beta amyloid model systems obtained from rigid docking calculations, and consisting of three different peptide structures—(i) K-peptide (54–62), (ii) STDY-K-peptide (50–62), and (iii) STDY-K-peptide at pH 2 (50–62-pH2), with the antiparallel (AP) and parallel (P) aggregation between neighboring beta strands—were then used as the starting structures for atomistic molecular dynamics (MD) simulations by the GROMACS 2019.6 package [52]. The explicit solvent simulations would take the aqueous environment, and protein dynamics, along with the in vitro temperature and pressure into account. A simulation setup started with defining a cubic simulation box, of a size around 6 × 6 × 6 nm^3^, which contained around 6000 water molecules when explicitly solvated. Proteins were parameterized through the CHARMM forcefield [53], and the TIP3P model was employed for all water molecules due to its compatibility with the protein forcefield [54]. For all simulation boxes, electric charge was neutralized by adding either Na + or C- counterions before a maximum 50,000-step energy minimization through the steepest descent algorithm. A simulated annealing MD run that increased the temperature from 100 K to 300 K within 1 ns was then performed, before a 100 ns productive MD run under an NPT ensemble with a temperature of 300 K, and pressure. The cutoff distance was set to 1.0 nm for short-range interactions, while the PME treatment was employed for the long-range electrostatics under periodic boundary conditions [55]. Temperatures and pressure were regulated by velocity rescaling [56] and Berendsen’s [57] algorithms, respectively, and P-LINC algorithm [58] was used to treat the holonomic constraints for all covalent bonds involving hydrogens, so that the simulation timestep of 2 fs was allowed.

### 3.3. Conformational and Interaction Analysis

After each simulation finished, the root mean square deviation (RMSD) of the backbone was calculated along the whole 100 ns trajectory, relative to the starting structures with all peptides perfectly aligned in either anti-parallel or parallel with their neighboring peptide chains. Per-residue root mean square fluctuation (RMSF) was also calculated from the last 50 ns of the trajectory for each residue of all 16 peptide chains, and was then averaged over the peptides. After that, the beta amyloid system was further characterized through geometrical parameters of polymer chains, i.e., time-averaged end-to-end distance for each strand over the last 50 ns, with radius of gyration (Rg) for the whole system of each beta amyloid calculated along the simulated trajectory. The order parameter, defined by *p* = 〈3 cos^2^ θ−1〉, was also calculated when θ was the relative orientation between any pair of vectors joining a C-alpha atom of residue i, and another C-alpha atom of residue i + 2 within the same strand. Finally, non-covalent interactions between different parts of each simulated beta amyloid system were analyzed through the calculations of molecular mechanics energy through the ‘g_mmpbsa’ suite [59]. Then, atom group indexing for each beta strand and sheet was defined through the ‘gmx make_ndx’ module. Total non-covalent interaction energy was calculated between each pair of 16 beta strands, and was presented in a 16 × 16 matrix through our in-house python script, in which the off-diagonal element represented pair interactions between strands, and the 8 × 8 off-diagonal submatrix represented pair interactions between sheets. Additionally, the interaction energy was decomposed into the contribution from each amino acid, and was averaged over all 16 beta strands. Visual Molecular Dynamics (VMD 1.9.3; University of Illinois Urbana-Champaign, Champaign, IL, USA) was the main software for MD trajectory visualization [60].

## 4. Conclusions

The synthesis of amyloid fibrils as functional biomaterials has become of interest. Extensive studies were carried out on model protein systems, including the hen egg white lysozyme (HEWL), for which several studies aimed to understand the fundamental aspect of this special type of protein aggregation. This computational study attempted to uncover the interplay between pH conditions, and molecular interactions underlying the formation of amyloid fibrils from short peptides contained within the HEWL. Atomistic molecular dynamics simulations of amyloid models, with 16 peptide strands, displayed greater model stability and compactness with antiparallel configurations, than with parallel configurations. Then, the bimodality of peptide aggregation mainly utilizing ‘electrostatics’ and ‘tryptophan’ was comparable to the pH-dependent amyloid aggregation of K-peptides and STDY-K-peptides experimentally investigated by Tokunaga et al. [43]. The formation of amyloid fibrils from K-peptides at both pH 2 and 7, observed in vitro, could be explained through the ordering of terminal residues and the energetic contribution of Trp62 residues. STDY-K-peptides could form a significantly lesser amount of amyloid fibrils under pH 7, in concurrence with the introduction of Asp52 residues. The negatively charged Asp52 interacted with Arg61 through the strong coulombic attractions between monopoles, which could disrupt the amyloid structure and decrease the order parameters. As the STDY-K-peptides were protonated and Asp52 became polar, uncharged residues, hydrophobic contacts, and the highly-ordered terminal residue arrangement became dominated by Trp62, which confirms a previous hypothesis [42]. This has helped us to visualize the atomistic details of terminal residues and the chain ordering of K-peptides and STDY-K-peptide for the first time. However, the more complex mechanisms of amyloid formation, from the entire amyloidogenic region of HEWL, still need further investigation.

## Figures and Tables

**Figure 1 ijms-24-02626-f001:**
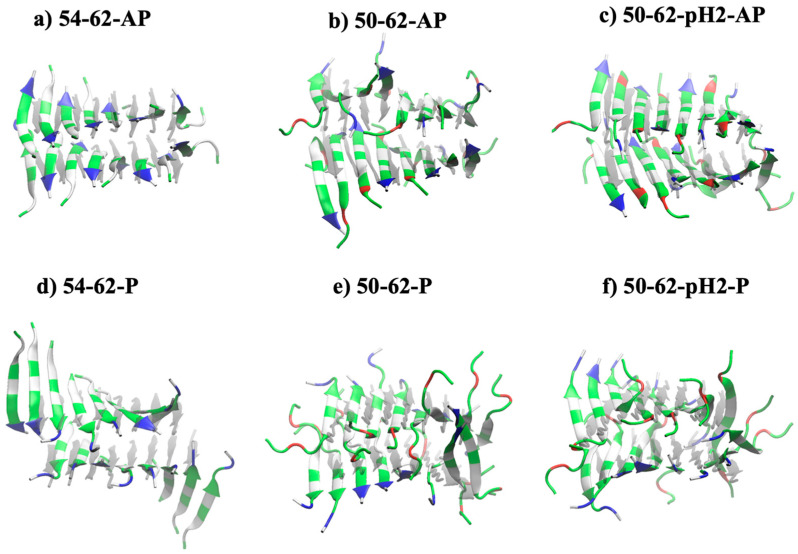
Conformational snapshots of simulated amyloid models after 100 ns MD simulations of (**a**) antiparallel model of K-peptide (54–62-AP), (**b**) antiparallel model of STDY-K-peptide (50–62-AP), (**c**) antiparallel model of protonated STDY-K-peptide at pH 2 (50–62-pH2-AP), (**d**) antiparallel model of K-peptide (54–62-P), (**e**) antiparallel model of STDY-K-peptide (50–62-P), (**f**) antiparallel model of protonated STDY-K-peptide at pH 2 (50–62-pH2-P). Blue color indicated positively charged residues, red color indicated negatively charged residues, green color indicated polar-uncharged residues.

**Figure 2 ijms-24-02626-f002:**
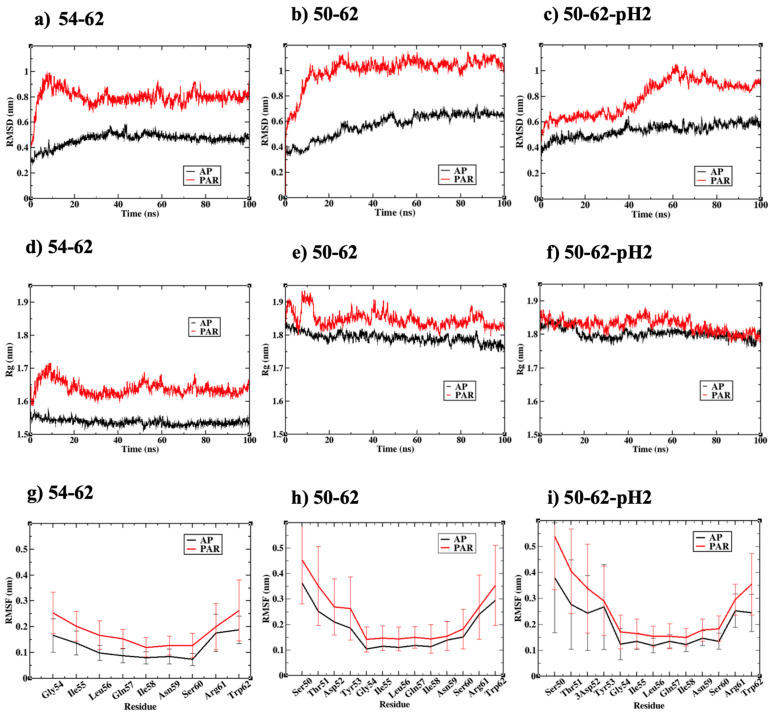
(**a**–**c**) root mean square deviation (RMSD) of all simulated systems along 100 ns trajectories, (**d**–**f**) radius of gyration (Rg) of all simulated systems along 100 ns trajectories, (**g**–**i**) root mean square fluctuation (RMSF) of all simulated systems calculated from the last 50 ns. Data presented in black annotated the antiparallel models and data presented in red annotated the parallel models.

**Figure 3 ijms-24-02626-f003:**
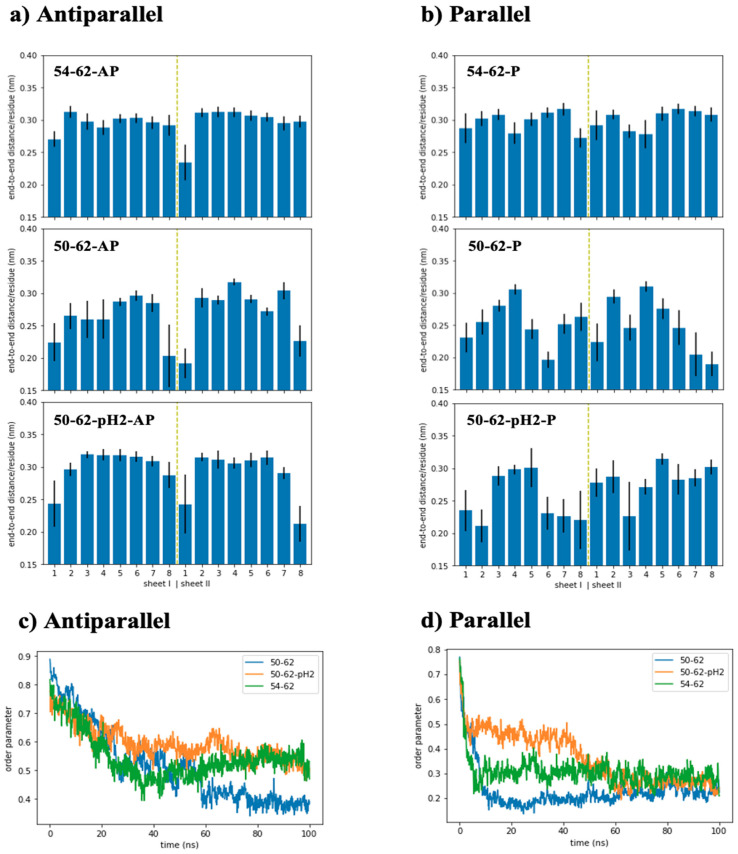
(**a**) time averaged end-to-end distance of each peptide strand from all the antiparallel amyloid models, (**b**) time averaged end-to-end distance of each peptide strand from all the parallel amyloid models, (**c**) order parameters calculated along 100 ns trajectories of all the antiparallel amyloid models, (**d**) order parameters calculated along 100 ns trajectories of all the parallel amyloid models.

**Figure 4 ijms-24-02626-f004:**
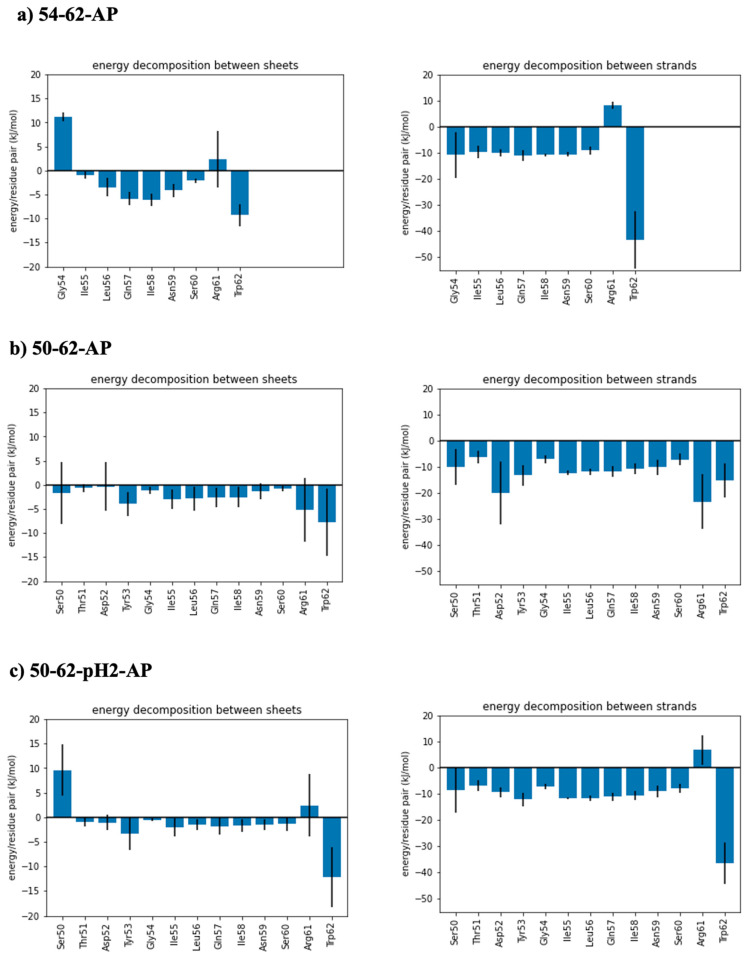
Time-averaged energy decomposition (left) between sheets and (right) between strands of (**a**) antiparallel model of K-peptide (54–62-AP), (**b**) antiparallel model of STDY-K-peptide (50–62-AP), and (**c**) antiparallel model of protonated STDY-K-peptide at pH 2 (50–62-pH2-AP).

**Figure 5 ijms-24-02626-f005:**
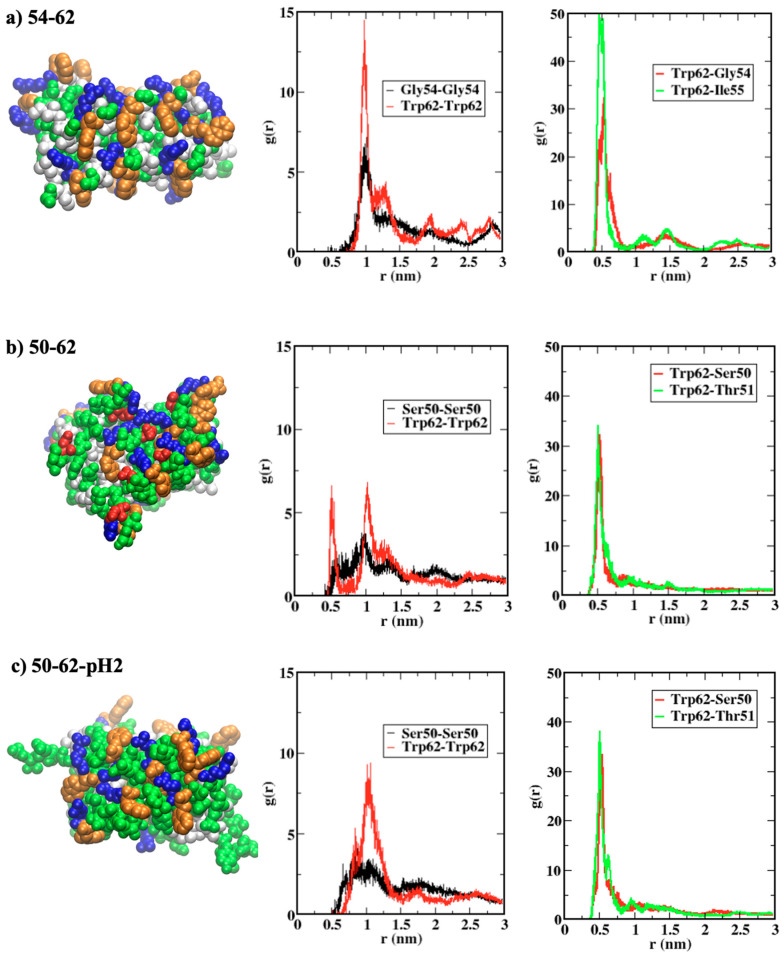
(**a**–**c**) (**left**) conformational snapshots after 100 ns showing terminal residues Trp62 in orange, positively charged residues in blue, negatively charged residues in red, polar uncharged residues in green, and hydrophobic residues in white, (**middle**–**right**) radial distribution functions (RDF) between pairs of terminal residues.

**Table 1 ijms-24-02626-t001:** Nomenclature and information of all simulations of beta amyloids in this study.

Name	Sequence	Asp Protonation	Orientation	No. of Strands	No. of Sheets
54–62-AP	GILQINSRW	N/A	Antiparallel	8	2
50–62-AP	STDYGILQINSRW	No	Antiparallel	8	2
50–62-pH2-AP	STDYGILQINSRW	Yes	Antiparallel	8	2
54–62-P	GILQINSRW	N/A	Parallel	8	2
50–62-P	STDYGILQINSRW	No	Parallel	8	2
50–62-pH2-P	STDYGILQINSRW	Yes	Parallel	8	2

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
