# Peer review of "Roles of Tryptophan and Charged Residues on the Polymorphisms of Amyloids Formed by K-Peptides of Hen Egg White Lysozyme Investigated through Molecular Dynamics Simulations"

_ijms, 2023, doi:10.3390/ijms24032626_

Round 1
Reviewer 1 Report
The authors performed molecular simulations of two short peptides derived from hen egg white lysozyme, namely, STDY-K-peptides and K-peptides. They found that anti-parallel configuration is the preferred configuration with greater stability. They also discovered some key residues that are important in fibrillation. To me, this work is well performed and is suitable for publication. Yet, some additional discussion or modification needs to be performed in the revised manuscript.
(1) I have to say that for short peptide, they tend to form antiparallel conformation in amyloid formation. This is a known experimental observation. It is not new. I think the authors should have some discussion on this issue to let the readers be award of this. Yet, using simulation to provide new theoretical insights into this issue is still valuable for understanding why short peptide tends to form amyloid with antiparallel configuration.
(2) The authors want to use the simulation work in this paper to give new insight into the amyloid formation of lysozyme. I have to say that lysozyme under heat and acidic condition will go through hydrolysis to generation a long sequence, such as 49-101. For long sequence, it usually forms parallel beta-sheet amyloid fibril like what we see in amyloid-beta and alpha-synuclein. The authors should include the hydrolysis information in the revised manuscript for the readers to better understand the fibrillation property of lysozyme. In other words, the work done with short peptide is not very relevant to understanding the actual fibrillation of lysozyme under pH=2 and 65 degree. Please refer to (Mishra R, Sorgjerd K, Nystrom S, et al. Lysozyme amyloidogenesis is accelerated by specific nicking and fragmentation but decelerated by intact protein binding and conversion[J]. J. Mol. Biol, 2007, 366 (3): 1029-1044). Yet, as I mentioned above, this work is still valuable for understanding why short amyloidogenic peptide tends to form amyloid with antiparallel configuration.
Author Response
Reviewer 1: The authors performed molecular simulations of two short peptides derived from hen egg white lysozyme, namely, STDY-K-peptides and K-peptides. They found that anti-parallel configuration is the preferred configuration with greater stability. They also discovered some key residues that are important in fibrillation. To me, this work is well performed and is suitable for publication. Yet, some additional discussion or modification needs to be performed in the revised manuscript.
Comment 1:
I have to say that for short peptide, they tend to form antiparallel conformation in amyloid formation. This is a known experimental observation. It is not new. I think the authors should have some discussion on this issue to let the readers be award of this. Yet, using simulation to provide new theoretical insights into this issue is still valuable for understanding why short peptide tends to form amyloid with antiparallel configuration.
Response: Thank you very much for the kind suggestion. We have now added ‘Starting structures included both parallel and antiparallel orientation of neighboring peptide pairs to further verify the known conformational stability of antiparallel beta amyloid systems.’ to the ending paragraph of our introduction
Comment 2:
The authors want to use the simulation work in this paper to give new insight into the amyloid formation of lysozyme. I have to say that lysozyme under heat and acidic condition will go through hydrolysis to generation a long sequence, such as 49-101. For long sequence, it usually forms parallel beta-sheet amyloid fibril like what we see in amyloid-beta and alpha-synuclein. The authors should include the hydrolysis information in the revised manuscript for the readers to better understand the fibrillation property of lysozyme. In other words, the work done with short peptide is not very relevant to understanding the actual fibrillation of lysozyme under pH=2 and 65 degree. Please refer to (Mishra R, Sorgjerd K, Nystrom S, et al. Lysozyme amyloidogenesis is accelerated by specific nicking and fragmentation but decelerated by intact protein binding and conversion[J]. J. Mol. Biol, 2007, 366 (3): 1029-1044). Yet, as I mentioned above, this work is still valuable for understanding why short amyloidogenic peptide tends to form amyloid with antiparallel configuration.
Response: We have now added ‘An earlier report on the low pH amyloid formation from HEWL at 65 ºC demonstrated the 49-101 region was amyloidogenic and the non-amyloidogenic 1-48 and 102-129 regions slowed down the formation [ref Mishra et al.]’ to our introduction and add ‘However, the more complex mechanisms of amyloid formation from the entire amyloidogenic region of HEWL still need further investigation.’ In our conclusions.

Reviewer 2 Report
The manuscript by Husnul Fuad Zein and Thana Sutthibutpong reports on the roles of charged residues and tryptophan residue on the amyloids formed by K-peptides from HEWL based on in silico analysis. The role of the influence of pH conditions and the molecular interactions underlying the formation of amyloid fibrils is also investigated. While the study is interesting, it has several concerns regarding in analysis and data representation.
1. In the Methodology the section 4.3 lacks data on program for visualizing three-dimensional structures shown in the figure 5, as well as a method (or program) for detecting the interaction of contacting amino acid residues between sheets of amyloid models of K-peptides.
2. It is worth changing parts 2.1 Disruption of the Antiparallel Beta Amyloid with a Negative Charge Addition and Table 1/Figure 1 by reformatting the text of the article. In the submitted version of article, the section Results begins with a table and a figure, which is difficult for the reader to understand.
3. Figure 2. Panels d-f must be made in the same scale as panels a-c (the abscissa has the same maximum value of 100 ns).
4. Figure 2. Panels g-i: root mean square fluctuation (RMSF) of all simulated systems calculate from the last 50 ns. 1) it is worth adding the name of amino acid residues (with the correct numbering as in the figure 4) instead of or next to the given numbering on the abscissa axis − this would really simplify the understanding of conclusions about the role of individual amino acid residues; 2) if I understand correctly, based on the given standard deviation interval for each point (number of individual amino acid residues), it cannot be argued that there is any definite difference between the curves for AP and PAR.
5. Please explain the reason for using the Avogadro software to model 3D structures of peptides instead of using special programs for predicting tertiary structures such as AlphaFold or I-Tasser, which according to the literature data, show the best results. Is modeling of protonated forms of peptides available in Avogadro software?
6. Have you tried to evaluate the role of individual residues in the stability of three-dimensional peptide structures with in silico screening of mutant forms of K-peptides, for example, using for example I-mutant or DeepDDG servers? There will probably be a correlation with your data on the role of Trp62.
Minor comments
Abstract lines 14, 15 - Using the same word “ previous” twice in one phrase. It would be better to rephrase.
Abstract lines 20, 21 – Using twice “STDY-K-peptides” in one phrase. Please, consider: “…STDY-K-peptides at pH 2 were higher 20 than those at pH 7.”
Lines 41, 51, 55, 65, 224, 314, 346 - in vitro and in silico should be written in italics
Line 51 - “used as a model protein to for the study” should remove “to”
Line 63 – “could promote amyloidosis of HEWL and was confirmed by a computational analysis [38]” should be corrected with “could promote amyloidosis of HEWL which was confirmed by a computational analysis [38]”
Line 85 – “The complete explanation on these different macroscopic outcomes of HEWL-based amyloids was still lacking”. “Bad” phrase “macroscopic outcomes” may be replaced with “features”.
Line 347 – nm3, 3 to top index
Line 367 – “After that, the simulated beta was further characterized” should be replaced with “beta amyloid system”
Author Response
Reviewer 2’s: The manuscript by Husnul Fuad Zein and Thana Sutthibutpong reports on the roles of charged residues and tryptophan residue on the amyloids formed by K-peptides from HEWL based on in silico analysis. The role of the influence of pH conditions and the molecular interactions underlying the formation of amyloid fibrils is also investigated. While the study is interesting, it has several concerns regarding in analysis and data representation.
Comment 1:
In the Methodology the section 4.3 lacks data on program for visualizing three-dimensional structures shown in the figure 5, as well as a method (or program) for detecting the interaction of contacting amino acid residues between sheets of amyloid models of K-peptides.
Response: We used VMD (Visual Molecular Dynamics) for visualizing three-dimensional structures and g_mmpbsa for detecting energy interaction of contacting amino acid residue between strands and sheets. G_mmpbsa. These are now mentioned in Section 4.3. We also provide more detail of sheet and strand pairwise interactions by adding these lines ‘Then, atom group indexing for each beta strand and sheet was defined through the ‘gmx make_ndx’ module. Total non-covalent interaction energy was calculated between each pair of 16 beta strands and was presented in a 16×16 matrix through our in-house python script, in which the off-diagonal element represented pair interactions between strands and the 8×8 off-diagonal submatrix represented pair interactions between sheets.’.
Comment 2:
It is worth changing parts 2.1 Disruption of the Antiparallel Beta Amyloid with a Negative Charge Addition and Table 1/Figure 1 by reformatting the text of the article. In the submitted version of article, the section Results begins with a table and a figure, which is difficult for the reader to understand.
Response: We moved Table 1 and Figure 1 below the text of part 2.1 and fixed the title accordingly.
Comment 3:
Figure 2. Panels d-f must be made in the same scale as panels a-c (the abscissa has the same maximum value of 100 ns).
Response: Those points in Figure 2 has now been fixed.
Comment 4:
Figure 2. Panels g-i: root mean square fluctuation (RMSF) of all simulated systems calculate from the last 50 ns. 1) it is worth adding the name of amino acid residues (with the correct numbering as in the figure 4) instead of or next to the given numbering on the abscissa axis − this would really simplify the understanding of conclusions about the role of individual amino acid residues; 2) if I understand correctly, based on the given standard deviation interval for each point (number of individual amino acid residues), it cannot be argued that there is any definite difference between the curves for AP and PAR.
Response: We named every residue in Figure 2. Panel g-i and we have removed the claims about the lower RMSF of AP configurations.
Comment 5:
Please explain the reason for using the Avogadro software to model 3D structures of peptides instead of using special programs for predicting tertiary structures such as AlphaFold or I-Tasser, which according to the literature data, show the best results. Is modeling of protonated forms of peptides available in Avogadro software?
Response: We needed straightened peptide as starting structures (similatly for all peptide sequences) in order to equally monitor their response to solvent environment after simulations start. We have added a phrase ‘to obtain straightened peptide as starting structures’ in Section 4.1.
Comment 6:
Have you tried to evaluate the role of individual residues in the stability of three-dimensional peptide structures with in silico screening of mutant forms of K-peptides, for example, using for example I-mutant or DeepDDG servers? There will probably be a correlation with your data on the role of Trp62.
Response: Thank you very much and it would be great to try. However, we cannot connect to DeepDDG server at the moment (there might be some server issues) and I-mutant did not accept short peptide predictions. If we could find some ways to test any of the in silico screening at any time before the final version, we will definitely add some lines to discuss them along with the results.
Minor Comments:
Abstract lines 14, 15 - Using the same word “ previous” twice in one phrase. It would be better to rephrase.
Response: Thank you very much for the kind suggestion. This error has now been fixed.
Abstract lines 20, 21 – Using twice “STDY-K-peptides” in one phrase. Please, consider: “…STDY-K-peptides at pH 2 were higher 20 than those at pH 7.”
Response: Thank you very much for the kind suggestion. This error has now been fixed.
Lines 41, 51, 55, 65, 224, 314, 346 - in vitro and in silico should be written in italics
Response: Thank you very much for the kind suggestion. This error has now been fixed.
Line 51 - “used as a model protein to for the study” should remove “to”
Response: Thank you very much for the kind suggestion. This error has now been fixed.
Line 63 – “could promote amyloidosis of HEWL and was confirmed by a computational analysis [38]” should be corrected with “could promote amyloidosis of HEWL which was confirmed by a computational analysis [38]”
Response: Thank you very much for the kind suggestion. This error has now been fixed.
Line 85 – “The complete explanation on these different macroscopic outcomes of HEWL-based amyloids was still lacking”. “Bad” phrase “macroscopic outcomes” may be replaced with “features”.
Response: Thank you very much for the kind suggestion. This error has now been fixed.
Line 347 – nm3, 3 to top index
Response: Thank you very much for the kind suggestion. This error has now been fixed.
Line 367 – “After that, the simulated beta was further characterized” should be replaced with “beta amyloid system”
Response: Thank you very much for the kind suggestion. This error has now been fixed.

Round 2
Reviewer 2 Report
The authors responded correctly to all my comments.